# A Practical Design Guide for Unbonded Jointed Plain Concrete Roads over Deteriorated HMA Roads: Realistic Traffic Loading

**Fatih İrfan Baş** [1,*]🆔, **Osman Ünsal Bayrak** [2]🆔 **and Halim Ferit Bayata** [1]🆔

1   Department of Civil Engineering, Faculty of Engineering and Architecture,
    Erzincan Binali Yıldırım University, Yalnızbağ Campus, 24002 Erzincan, Turkey
2   Department of Civil Engineering, Faculty of Engineering, Ataturk University, 25240 Erzurum, Turkey
*   Correspondence: fibas@erzincan.edu.tr

**Abstract:** This study aimed to determine the material properties and the thickness of the layers of an unbonded jointed plain concrete pavement design over a deteriorated flexible pavement that lost its service capability. Maximum principal stresses occurring under the concrete pavement should not exceed the modulus of rupture of the concrete. Three-dimensional finite element analyses using ANSYS (Release 18.1 SAS IP, Inc.) software were carried out with the Taguchi method. The most reliable solution for the pavement design was investigated with whole configuration axle loading that reflects the realistic traffic situation. The stress values under the concrete slab due to the positive temperature gradient neglected in the American Association of State Highway and Transportation Officials (AASHTO) method were also investigated. The concrete slabs are exposed to these stresses for longer than the axle loads. Temperature distributions throughout the thickness of the concrete slab were calculated with the bilinear formulas suggested in the study. Axle loadings were applied on both the pavement edge and corner to reflect the most critical loading condition in the pavement design. The critical axle type was found to be 1.2. It was observed that a 0.15 m concrete thickness was appropriate for the concrete class, joint spacing and hot mix asphalt (HMA) elasticity modulus levels used for the 1.22, 1.122, and 1.2 + 111 axle types, but it was not appropriate for the 1.2 axle types. The slab thicknesses calculated with the AASHTO method, and the finite element method were found to be close to each other. It was determined that the concrete thickness significantly affected the maximum principal stress, with a performance statistic (S/N) value of 1692 for the 1.2 axle type. This was followed by the modulus of elasticity of the concrete with an S/N value of 0.356, the modulus of elasticity of the existing flexible pavement with an S/N value of 0.244, and the concrete joint spacing with an S/N value of 0.105. A practical design guide was recommended to extend the service life of a highly deteriorated flexible pavement with the construction of unbonded jointed plain concrete.

**Keywords:** jointed plain concrete pavement; deteriorated HMA pavement; finite element method; Taguchi method; AASHTO

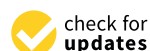



## 1. Introduction

Road pavement design includes determining the pavement and slab thicknesses and calculating the traffic load that the pavement is able to carry safely without being exposed to deformations throughout the project life, considering the changes in climate, traffic, and material conditions [1].

Three types of flexible, rigid, and composite pavements have been used. Flexible pavements include a layered structure with surface, binder, base, and subbase layers, and rigid pavements include concrete slabs built on the subgrade [2]. On the other hand, composite pavements are obtained by constructing a hot bituminous mixture (HMA) layer on deteriorated concrete pavements, or concrete pavement on deteriorated flexible pavements [3].

The most frequently used rehabilitation method is asphalt pavement on HMA pavement. Milling may be required depending on the HMA pavement condition [4]. A new HMA pavement layer can be used to eliminate functional distresses in general and design-dependent structural problems [5]. However, the performance of the new HMA pavement depends on the condition of the subbase. One study, examining the performance of HMA pavement constructed on HMA pavement for rehabilitation [6], revealed that rutting and cracks in the underlying layer could appear again in the newly constructed pavement.

In recent years, despite the declining budgets, road engineers have made great efforts to focus on successful preservation strategies for road networks regarding pavement sustainability and protection due to increasing traffic volumes and loads [7]. In various pavement preservation activities, resurfacing an existing pavement with concrete pavement is a less costly, quick, and sustainable solution compared to a complete reconstruction [8].

Improving HMA pavement with Portland cement concrete (PCC) has many advantages; for example, improving skid resistance and road safety (especially under wet conditions) is more advantageous than constructing HMA pavements [9]. Loss of time and delays caused by periodic maintenance of asphalt pavements have considerably decreased. The proven durability and long-term performance of a PCC surface reduce the maintenance duration and life-cycle cost of HMA pavements [9]. This has been supported by studies on concrete pavement projects in the USA state of Nebraska [10].

Whitetopping (WT), defined as a Portland Cement Concrete (PCC) pavement overlay on HMA pavement, can be classified according to the thickness and type of bond to the underlying HMA layer [11]. When considering the concrete overlay thickness, there are three types of concrete overlay conventional (WT) (thickness > 200 mm), thin whitetopping (TWT) (100–200 mm thickness), and ultra-thin whitetopping (UTW) (50–100 mm thickness) [12].

In previous studies on concrete and flexible pavement design, Sultana [13] evaluated the behavior of thin concrete pavements on HMA pavements for different bonding conditions considering pavement thicknesses, shoulder conditions, HMA moduli of elasticity, and temperature differences. Using the Solid Works finite element program, the author modeled the three-layer road system consisting of a pavement, an existing flexible pavement, and natural soil. A single axle with dual tire loading was applied on the pavement. It was stated that with an increase of 50 mm in the HMA thickness, the concrete tensile stresses decreased by about 12%–14%. Zbiciak et al. [14] investigated the effects of loads due to temperature differences and single-wheel loads on concrete pavement using the finite element method. It was stated that an increase in the coating thickness causes stress reduction. Salman and Patil [15] developed a 3D finite element model for the investigation of stresses due to temperature differences in concrete pavements. Çelik [16] developed a 3D finite element model consisting of a concrete pavement layer, sub-base layer, and ground layer to analyze stresses due to tandem-axle dual tire loading. It was stated that the stresses arising under the concrete pavement were vertically 119% and horizontally 74% less than those of the flexible pavement. Kumara et al. [17] developed a 3D finite element model for the stress analysis of UTW pavements under critical loading conditions to analyze UTW test pavement sections for Florida conditions. The UTW sections in poor condition were found to have more tensile stresses under critical loading in the finite element model. Mustaque and Sharmin [18] developed a 3D finite element model to investigate the behavior of concrete pavement applied on deteriorated HMA pavement. Considering a single axle with dual tire loading, different layer thicknesses, different asphalt concrete elasticity moduli, the bond status between the two layers, and the stresses occurring in the concrete pavement were examined. The study revealed that the bond status between the asphalt and concrete pavement is the most critical factor affecting concrete pavement behavior. It has been stated that the tensile stresses in concrete pavement decrease with the increase in concrete and asphalt coating thicknesses. Hu and Walubita [19] developed a 3D finite element model to investigate the bonding condition between the pavement layers. The bonding condition significantly affected the tensile, compressive, and shear stresses/strains

in the AC pavement structures. Williamson [20], using the ABAQUS program, investigated the effect of the adhesion layer applied on an asphalt pavement made for improvement of an existing asphalt pavement. Single axle dual tire loading was applied on the pavement. It was revealed that the bond status between the newly built asphalt layer and the existing asphalt layer significantly affects the tensile stresses under the freshly built layer. Ozer et al. [21] conducted finite element and field experiments to investigate the effect of the bond status between the asphalt and concrete pavement. A moving load with a dual-tire assembly and a wide-base tire was applied to the pavement. It was stated that the impact of the bond state on pavement responses increased as the pavement temperature increased. A non-linear finite element model was developed to analyze continuous RC beams having web openings strengthened with different schemes of fiber-reinforced polymers [22]. It was stated that the reduction in load capacities ranged from 7.3% to 66.1% compared to the solid beam. Baraghith et al. [23] conducted finite element analyses and field experiments to investigate strain-hardening cementitious composite strips internally reinforced with glass fiber textile mesh layers for the shear strengthening of RC beams. It was concluded that the strengthened RC beams' shear carrying capacity was 47%–142% higher than that of the control specimens. Sakr et al. [24] investigated the behavior of concrete plates strengthened in shear with ultra-high performance fibers. It was reported that the plate case with strengthening on one side prevented shear cracks from appearing on the reinforced side. Hamoda et al. [25] examined the behavior of steel I-beams with high-strength bolted connectors embedded in steel fiber-reinforced concrete. A finite element model was developed and compared with the experimental results. A formula was proposed to estimate the ultimate shear bearing capacity with the push-out failure mode. A finite element model was developed to compare experimental and numerical results of the bonding behavior between concrete and prefabricated ultrahigh-performance fiber-reinforced plates [26]. A design equation was proposed to estimate the bond stress between the concrete and prefabricated plates. Mansour and Fayed [27] assessed the effects of surface preparation methods on the bond performance of composite plates. It was concluded that the cohesion of all composites with the proposed preparation methods was within acceptable limits. Basha et al. [28] investigated flexural strengthening of slabs with strain-hardening cementitious composites. A 3D finite element model was developed to compare the results with field tests. A theoretical equation was proposed for predicting the ultimate load of the strengthened slabs. Al Harki et al. [29] developed a finite element model with tandem-axle dual tire loading to assess the influence of hybrid steel fiber on the behavior of rigid pavements. They concluded that hybrid fiber reinforcement greatly enhanced the strength and flexibility of the pavements.

### 1.1. Visual Evaluation of HMA Pavement

Visual evaluation is the process of evaluating the pavement condition as good, fair, poor, or deteriorated, as shown in Figure 1. When in a good condition, the pavement is structurally sound, but there is a need to increase structural capacity, improve surface properties, and remove surface defects. When in a fair condition, the pavement is structurally sound but has minor surface distresses such as potholes, blocks, and thermal cracking. When in a poor condition, the pavement has alligator cracks, rutting, shoving, base-to-surface and surface-to-base aggregate stripping, and usual material-dependent problems. When in a deteriorated condition, the pavement has a poor condition and significant surface deterioration, surface-to-base and base-to-surface aggregate stripping, thermal expansion, and structural distress [7].

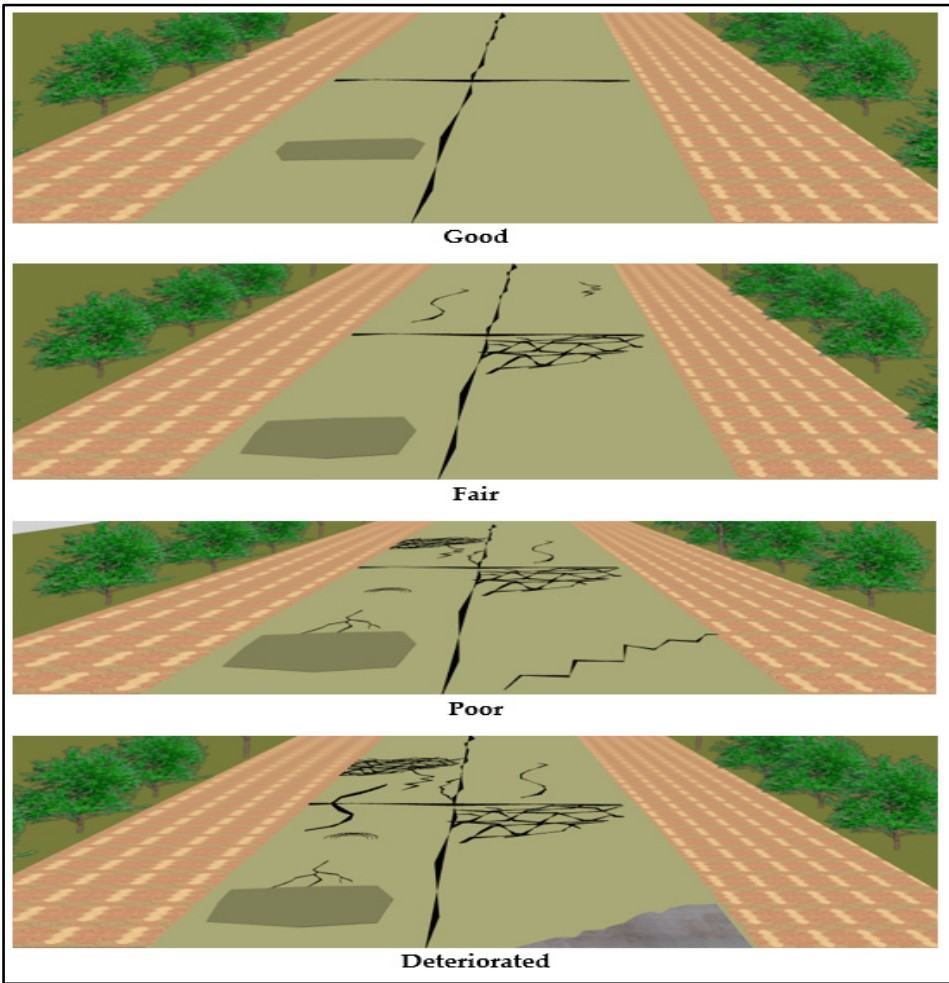

**Figure 1.** Visual evaluation of HMA pavement. Adapted/Redrawn from Ref. [7].

### 1.2. Research Significance

The main objective of the current study is to numerically investigate the extension of the service life of an HMA pavement that lost its serviceability due to structural and functional deterioration with an unbonded concrete overlay construction. The maximum principal stresses that occur under the unbonded concrete overlay were estimated using finite element modeling with the Taguchi method. Studies in the literature and mathematical formulas proposed by Westergaard and Bradbury were used to ensure the finite element models' efficiency.

The recent technology for constructing HMA pavement has become increasingly unsustainable due to increasing life cycle costs. For that reason, rigid pavement construction should be preferred worldwide and in our country. The load bearing capacity of an HMA pavement structure that is no longer usable, or losing its functional properties, is often higher than that of a typical base course. Therefore, reusing such a pavement structure reduces the thickness of the other pavement layers eliminating the labor-intensive and costly replacement of the base layer.

Whereas the administration of bonded or unbonded concrete overlay on HMA has increased rapidly worldwide, they have not yet been administered in our country. Constructing a concrete overlay designed to increase the structural capacity and eliminate surface problems on an HMA pavement in good or fair condition, which is structurally sound but has surface problems such as smoothness, friction, and noise, will not be actualized in our country under today's needs. For that reason, the deteriorated condition (severe surface deterioration, aggregate stripping, thermal expansion, and structural problems),

namely, the unbonded design, was used for the HMA pavement in the finite element analyses performed depending on rural roads.

In contrast to Westergaard's assumption of a linear temperature distribution, temperature profiles measured at the depth of concrete slabs are not linear. In concrete pavements, tensile stresses occur under the slab and compressive stresses occur above the slab due to positive temperature differences. Whereas the exposure time of the tensile stresses under the concrete slab caused by the axle loads is short, the exposure time of the tensile stresses caused by the temperature differences is longer. Therefore, these tensile stress values, due to the positive temperature gradient formed under the slab, which are efficient for longer compared to axle loads, should be specifically investigated. The AASHTO method regards the roadbed soil resilient modulus (MR), which constantly changes throughout the year depending on frost sensitivity instead of non-linear temperature distributions occurring throughout the concrete slab thickness in concrete overlay designs constructed on HMA pavements. However, a more realistic approach includes these tensile stresses caused by nonlinear temperature distributions under the slab that are neglected in the AASHTO method for pavement design.

In previous studies in the literature, the loadings for HMA pavement and rigid pavement models created using finite element programs were administered as single, tandem, or tridem axle. This does not fully reflect the actual loading situation on highways. Therefore, it is a more accurate approach for a pavement design to be actualized with actual loading conditions of en-route vehicles on roads, with whole configuration axle loading. This study aimed to create loading models using the most frequent en-route heavy vehicle types on our country's highways.

This study aimed to determine the concrete class, slab thickness, and dimensions using finite element analysis (ANSYS), and to compare them with the AASHTO method to extend the service life of an HMA pavement that lost its serviceability due to structural and functional deterioration and provide the maximum principal stress possible under an unbonded concrete overlay to be constructed under the defined design criteria (tensile modulus of concrete). The most critical loading conditions on the slab were used for concrete pavement design, edge and joint loading, whole configuration axle type loading, and temperature distribution values changing throughout the concrete slab thickness.

This study aimed to analyze the relationship between the concrete class, modulus of elasticity of HMA, modulus of elasticity of concrete, joint spacing, and maximum principal stress, and to create prediction models with the Taguchi method using the results obtained from the finite element analysis.

## 2. Materials and Methods

### 2.1. Verification of the ANSYS Model Established with Literature Studies

2.1.1. Verification of Stresses Induced by Temperature Difference

Salman and Patil [15] developed a 3D finite element model for investigating the stresses induced by temperature differences in concrete pavements. For this, the ANSYS structural analysis package was used. The model established in ANSYS was verified using Salman and Patil's [15] model parameters. The maximum stress caused by the linear temperature distributions administered to the concrete pavement using positive and negative gradients is presented in Figures 2 and 3, respectively.

Salman and Patil [15] found the maximum stress in the concrete pavement due to a positive temperature gradient of 0.66 °C/cm to be 1.809 MPa, and that due to a negative temperature gradient of −0.33 °C/cm to be 0.897 MPa. The established ANSYS verification model calculated the resulting temperature stresses as 1.808 MPa and 0.899 MPa, respectively.

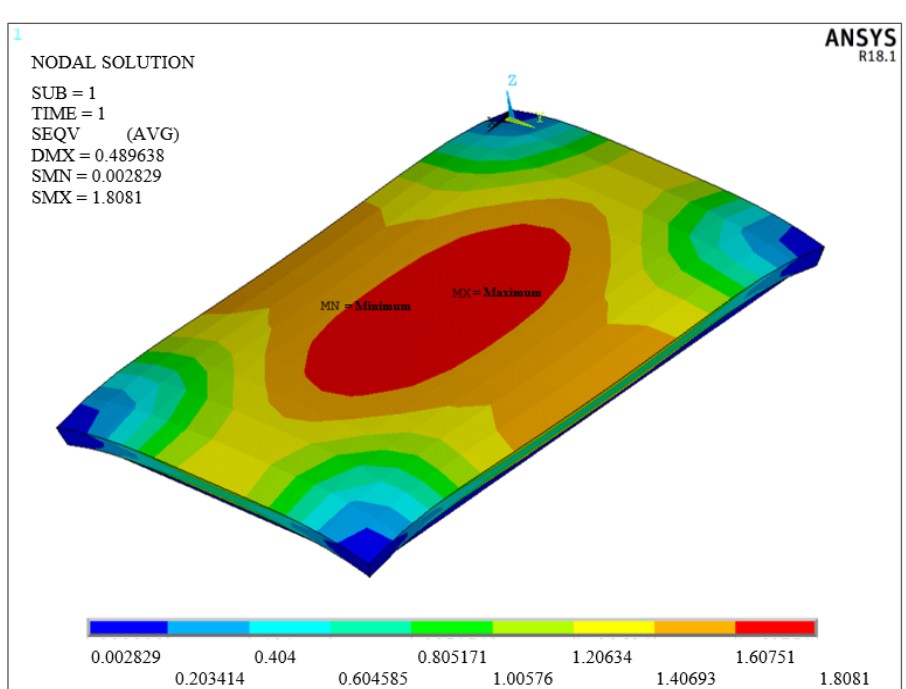

**Figure 2.** Maximum stresses induced by a positive gradient (ANSYS R18.1).

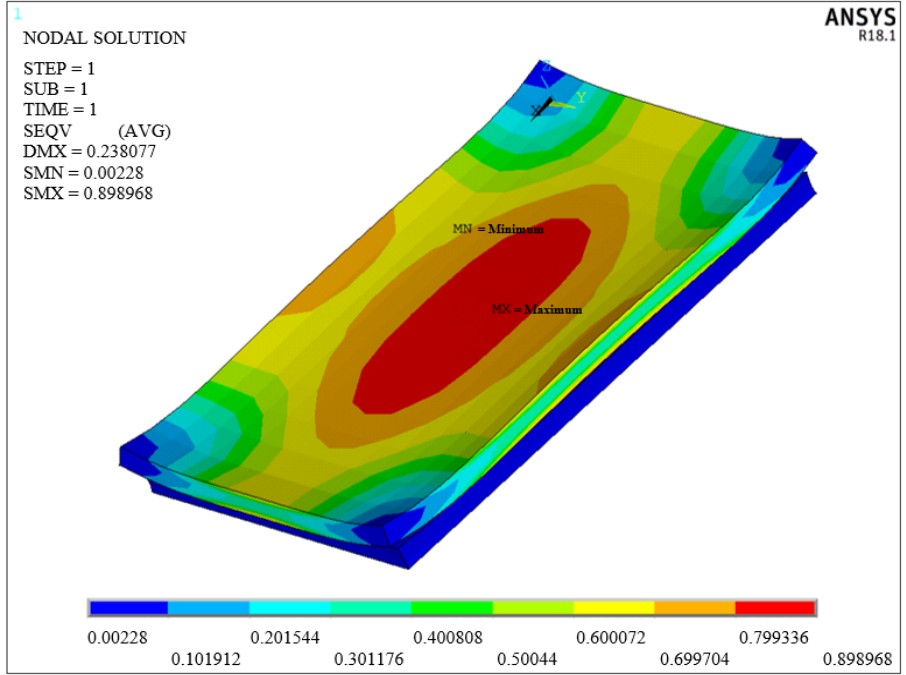

**Figure 3.** Maximum stresses induced by a negative gradient (ANSYS R18.1).

### 2.1.2. Verification of Stresses Induced by Single Wheel Loads

Çelik [16] developed a 3D finite element model including concrete pavement, subbase, and soil layers to analyze stresses induced by single wheel loads. The ANSYS structural analysis package was used for this. The model established in ANSYS was verified using the model parameters of [16].

The minimum and maximum stresses in the Y direction are presented in Figure 4. The minimum and maximum stresses were calculated to be −1.249 MPa and 0.202 MPa, respectively.

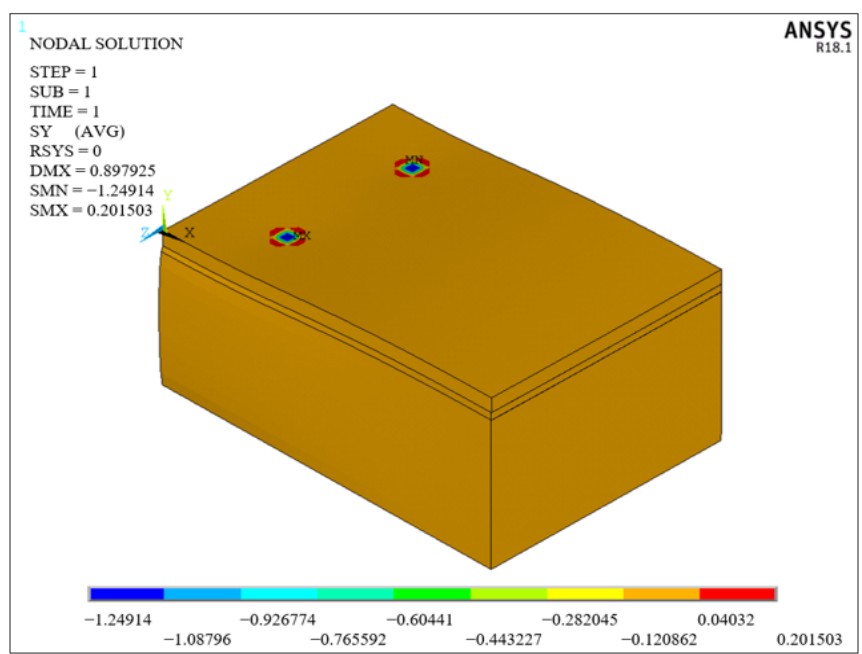

**Figure 4.** The minimum and maximum stresses in the Y direction (ANSYS R18.1).

In Figure 5, the minimum and maximum stresses in the X direction are presented. The minimum and maximum stresses were calculated to be −1.864 MPa and 1.488 MPa, respectively.

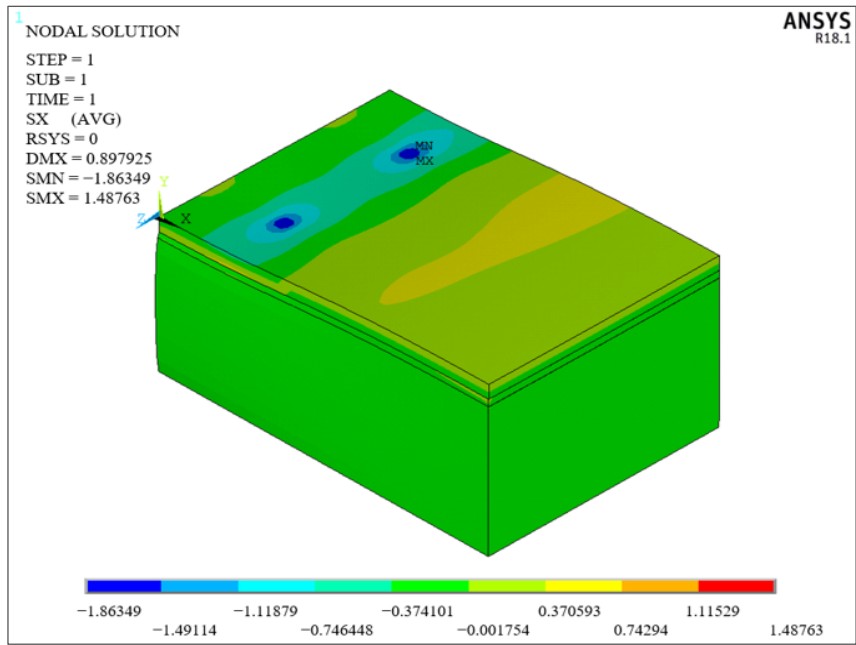

**Figure 5.** The minimum and maximum stresses in the X direction (ANSYS R18.1).

The comparison of the stress and strain calculated with the verification model created using the results of [16] is presented in Table 1. The verification model obtained a good correlation ($R^2$ of 0.982).

**Table 1.** Comparison of the wheel loads verification model.

| The Study | Total Deformation mm | Deformation in Y Direction mm | | Deformation in X Direction mm | | Tension in Y Direction MPa | | Tension in X Direction MPa | |
|---|---|---|---|---|---|---|---|---|---|
| | DMX | SMN | SMX | SMN | SMX | SMN | SMX | SMN | SMX |
| Çelik [16] | 0.889 | −0.889 | 0.090 | −0.038 | 0.143 | −1.219 | 0.301 | −1.823 | 1.039 |
| ANSYS Verification Model | 0.898 | −0.898 | 0.081 | −0.037 | 0.122 | −1.249 | 0.202 | −1.864 | 1.488 |

*2.2. Calculation of Unbonded Concrete Pavement Thickness on HMA Pavement with AASHTO Method*

The subgrade was removed from the calculation to see the amount of displacement in the finite element analysis, and the bottom of the base course layer was taken as the fixed support [30]. To meet the same conditions, the effective dynamic k-value was taken as 1000 PCI, and the effective static k-value was taken as 500 PCI [31]. The concrete overlay thicknesses to be constructed on the flexible pavement calculated for the concrete classes of C20/25 to C50/60 with the AASHTO method are presented in Table 2.

**Table 2.** Concrete pavement thicknesses calculated for concrete classes of C20/25 to C50/60 with the AASHTO method.

| Concrete Class | C20/25 | C25/30 | C30/37 | C35/45 | C40/50 | C45/55 | C50/60 |
|---|---|---|---|---|---|---|---|
| 18 kip (8,2 t) $W_{8,2}$ | $3 \times 10^6$ | $3 \times 10^6$ | $3 \times 10^6$ | $3 \times 10^6$ | $3 \times 10^6$ | $3 \times 10^6$ | $3 \times 10^6$ |
| R | 95% | 95% | 95% | 95% | 95% | 95% | 95% |
| $Z_R$ | −1.645 | −1.645 | −1.645 | −1.645 | −1.645 | −1.645 | −1.645 |
| $S_0$ | 0.35 | 0.35 | 0.35 | 0.35 | 0.35 | 0.35 | 0.35 |
| $P_t$ | 2 | 2 | 2 | 2 | 2 | 2 | 2 |
| $P_0$ | 4.5 | 4.5 | 4.5 | 4.5 | 4.5 | 4.5 | 4.5 |
| Modulus of rupture of PCC—$S_c$ (psi) | 464 | 522 | 551 | 609 | 638 | 667 | 725 |
| $C_d$ | 1 | 1 | 1 | 1 | 1 | 1 | 1 |
| Load transfer coefficient—J | 3.8 | 3.8 | 3.8 | 3.8 | 3.8 | 3.8 | 3.8 |
| Modulus of elasticity of concrete—$E_c$ (psi) | $4.06 \times 10^6$ | $4.35 \times 10^6$ | $4.64 \times 10^6$ | $4.79 \times 10^6$ | $4.93 \times 10^6$ | $5.22 \times 10^6$ | $5.37 \times 10^6$ |
| Modulus of subgrade reaction—k (PCI) | 500 | 500 | 500 | 500 | 500 | 500 | 500 |
| $\log_{10} W_{8,2}$ | 6.477 | 6.477 | 6.477 | 6.477 | 6.477 | 6.477 | 6.477 |
| Overlay thickness—D (inch) | 10.44 | 9.78 | 9.51 | 8.97 | 8.73 | 8.54 | 8.12 |
| Overlay thickness—D (m) | 0.265 | 0.248 | 0.242 | 0.228 | 0.222 | 0.217 | 0.206 |

*2.3. Finite Elements Model*

2.3.1. Axle Configurations

Axle types used in heavy vehicles worldwide are presented in Figure 6. When the distribution of axle types according to vehicle classes obtained from the axle weight studies carried out at 109 points by the General Directorate of Highways considering 20,917 vehicles throughout Turkey was analyzed, it was noticed that the three-axle (1.22), two-axle (1.2), and four-axle (1.122) trucks were driven on the roads at most with rates of 36%, 27%, and 16%, respectively. Five-axle (1.2 + 111) semi-trailers were driven on the roads at most with a ratio of 98% [32]. Therefore, the axle types specified in the finite element analysis were used.

When the weight data according to the axle types were analyzed, it was noticed that the average heavy vehicle total weights were highly below the actual legal weight limits [32]. Although the data obtained from the studies had average values, while some trucks were within the legal weight limits, some trucks were en-route with no load or less load, so the total legal weights in force in Turkey were used while performing the finite element analysis. However, instead of the legal loading limit of 11.5 tons for the front axles of trucks and tow trucks, the technical load value of 8.2 tons was used.

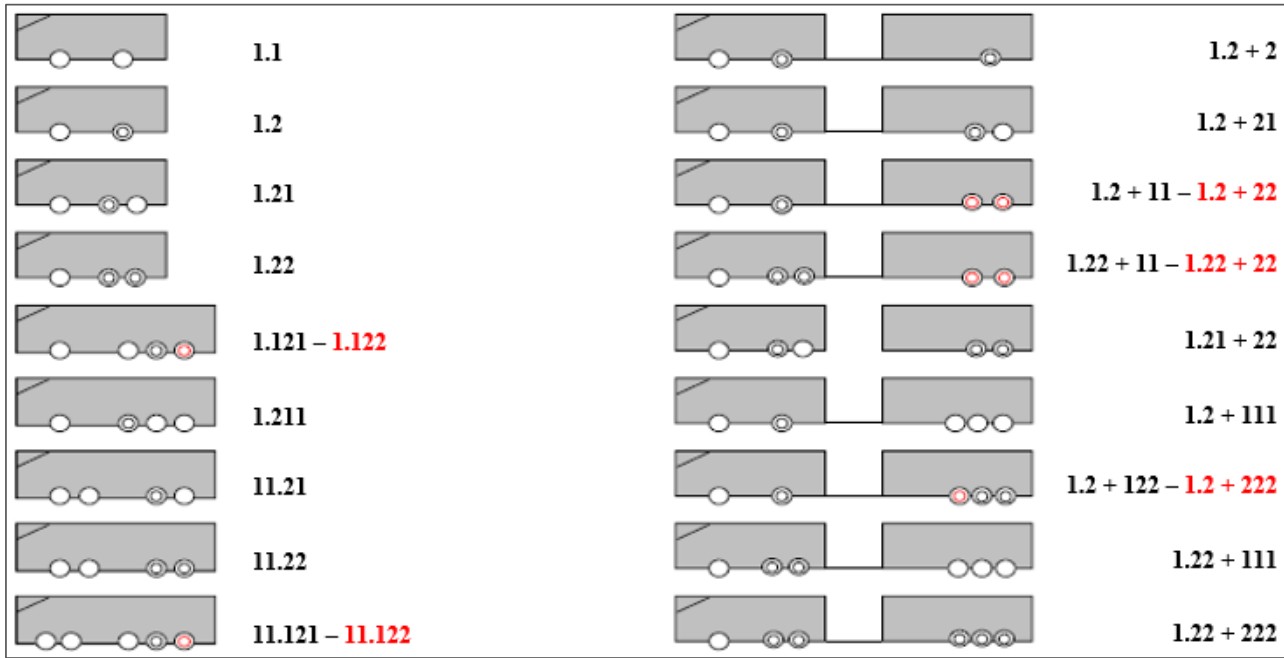

**Figure 6.** Axle types used in heavy vehicles.

Pavement design methods such as AASHTO were developed considering the road test results and the effect of the standard axle load absolute frequency on pavement performance. In the HMA and rigid pavement models modeled with finite element analysis in previous literature studies, the loading was administered as single-axle, tandem-axle, or tridem-axle. However, this did not entirely reflect the actual loading on the road. Therefore, in this study, whole configuration axle loading was conducted. In Table 3, the wheelbases are presented for en-route trucks and lorry tow trucks of various trademarks traveling on our country's highways and worldwide. In this study, although the brand and wheelbase of the en-route vehicles were unknown, modeling was established by taking the shortest wheelbase in terms of critical loading among the vehicle brands according to the number of vehicle axles.

**Table 3.** Wheelbases used in various commercial brands.

| Type of Vehicle | Type of Axle | Wheelbase (mm) | | | | | |
|---|---|---|---|---|---|---|---|
| | | Mercedes | Ford | Man | Volvo | DAF | Renault |
| Two-axle truck | 1.2 | 3300 to 6600 | 4750–5500 | 3900 to 5100 | 3400 to 6700 | 3450 to 7300 | 3400 to 6700 |
| Three-axle truck | 1.22 | 3600 to 5700 | 4460–5410 | 3300 to 5100 | 3885 to 6685 | 4500 to 6600 | 3500 to 6000 |
| Four-axle truck | 1.122 | 3600 to 5700 | 5100–5200–5900 | 3600 to 5700 | 4300 to 5885 | 4700–5300 | 4300 to 6400 |
| Five-axle semi-trailer | 1.2 + 111 | 3300–3600–3900 | 3600 | 3600–3900 | 3500–3600–3700–3800 | 3600–3800 | 3500–3700–3800–3900 |

The modeled loading conditions for trucks with axle types of 1.2, 1.22, 1.122, and 1.2 + 111 in the finite element analyses are presented in Figure 7a–d, respectively. The dimensions are given in mm.

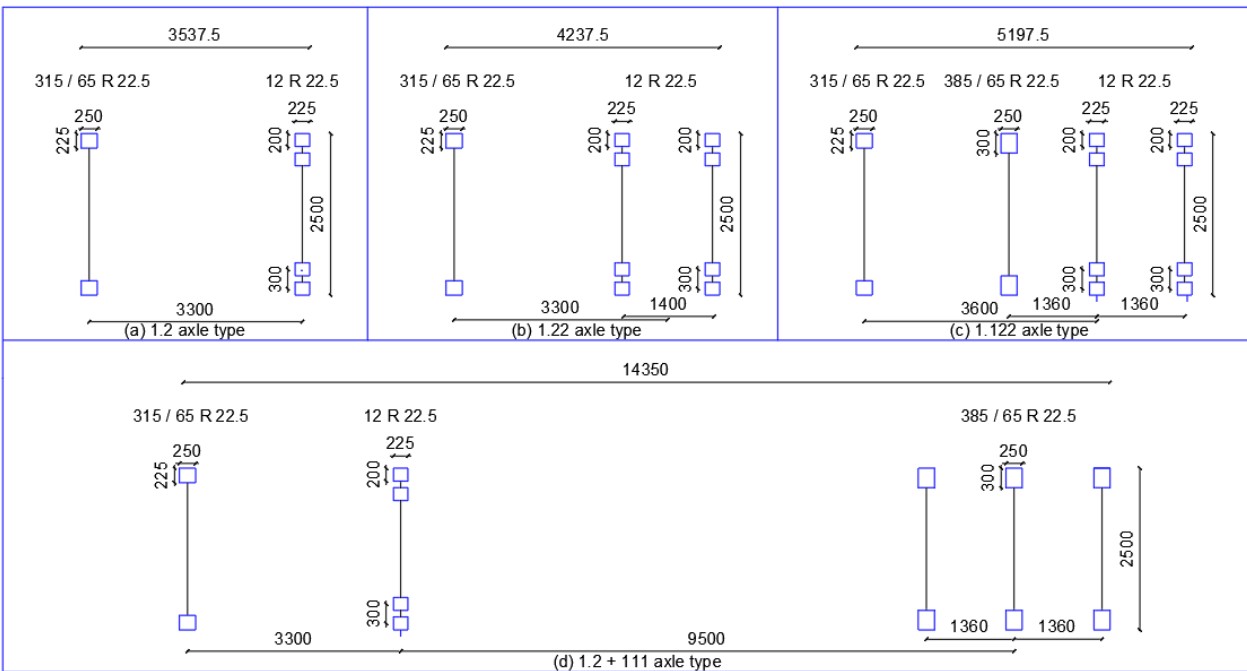

**Figure 7.** Modeled loading situations.

2.3.2. Tire–Pavement Contact Areas and Inflation Pressure

The most used tire sizes and classes in heavy commercial vehicles which were obtained from Michelin's Turkey-wide sales data [33]. Briefly, 315/80 R 22.5 class tires for front wheels and 12 R 22.5 class tires for dual rear wheels were used on rigid trucks for whole configuration finite element modeling under these data. Meanwhile, 315/80 R 22.5 class tires were used for tow trucks and semi-trailers, 12 R 22.5 class tires were used for dual wheels, and 385/65 R 22.5 class tires were used for single wheels on semi-trailers.

Moffatt [34] reported that the tire pressure in heavy vehicles crossing the roads ranges from 500 kPa to 1000 kPa and recommended 700 kPa and 800 kPa tire pressures for structural design. Korkiala-Tanttu [35] measured tire contact areas with accelerated road tests and laboratory studies for dual and wide-type single wheels under 500, 600, 700, 800, and 900 kPa tire pressures. In this study, the inflation pressure of all tires was 700 kPa, which is the average wheel inflation pressure value. For the tire contact areas to be used in the finite element model, the areas found by Korkiala-Tanttu [35], using accelerated road tests and laboratory studies for dual and wide-type single wheels under different wheel loads and tire pressures, were used [33]. The contact areas and axles used in the whole configuration modeling are presented in Table 4.

**Table 4.** Modeled tire types and contact areas.

| Type of Tire | Wheel Load (kg) | Tire Inflation Pressure (kPa) | Tire Contact Area Length (mm) × Width (mm) | Placement |
|---|---|---|---|---|
| 315/80 R 22.5 | 4100 | 700 | 250 × 225 | Front Tires on Trucks and Tow Trucks |
| 385/65 R 22.5 | 4100 | 700 | 250 × 300 | Tires on Semi-Trailers |
| 12 R 22.5 | 4100 | 700 | 225 × 200 + 225 × 200 | Dual Tires |

2.3.3. Pavement Layer Thicknesses and Material Properties

The yield and fracture criteria determine whether the material is damaged against an applied load. Whereas ductile materials stretch before breaking, brittle materials suddenly break before stretching. Von Mises stress is used as the yield criterion for ductile materials. Maximum principal stress is used as the yield criterion for brittle materials. When the max-

imum principal stress is equal to the yielding or rupture strength of the material, yielding or rupture appears in the material [36]. Shoukry et al. [37] concluded that the maximum principal stress is a better indicator of the overall stress in the pavement, especially when considering the combined effects of temperature and axle loads. For that reason, the finite element analysis used maximum principal stress to evaluate the tensile stresses under the concrete slab.

C40/50, C45/55, and C50/60 were used as rigid pavement concrete classes. The concrete rupture modulus was taken to be 4.4 MPa, 4.6 MPa, and 5 MPa, respectively [38].

The modulus of elasticity of the existing HMA pavement is one of the necessary inputs for structural analysis. The properties of the HMA layer can be characterized using a falling weight deflectometer (FWD); however, the temperature during the test ultimately affects the values of the back-calculations because the modulus of elasticity of the HMA pavement material can change significantly throughout the year. Therefore, an average value should be used. It has been suggested that the modulus of elasticity is taken to be approximately 690 MPa for an old HMA pavement in deteriorated condition (severe fatigue cracks), about 2413 MPa for an HMA pavement in a fair condition (some structural problems), and about 4136 MPa for an HMA pavement in a good condition (surface problems such as rutting that can mostly be eliminated by milling alone) [39,40]. In this study, the modulus of elasticity of the existing asphalt layer was taken as 690, 1379, and 2068 MPa, the unit weight was taken as 2400 kg/m$^3$, and the Poisson ratio was taken as 0.3.

According to AASHTO, the modulus of elasticity of granular material ranges from 103 to 310 MPa. In this study, to reflect the worst case, the modulus of elasticity for the base layer under the HMA pavement was 103 MPa. The unit volume weight was taken as 2100 kg/m$^3$. The Poisson ratio was taken as 0.3.

According to the Flexible Pavement Project Planning Guide of Turkey General Directorate of Highways, roads with an equivalent single axle load of less than 3 million are low volume. The AASHTO method gives rigid plate thicknesses up to 1 million equivalent single axle loads (ESALs) for low-volume roads. In this study, to reflect the deteriorated condition, the ESAL was taken to be 3,000,000.

AASHTO revealed the minimum HMA and base layer thicknesses according to the ESAL. In this study, the HMA layer thickness was regarded as 87.5 mm, and the base layer thickness was considered to be 150 mm. Concrete pavement thicknesses were considered to be 100 mm, 150 mm, and 200 mm.

### 2.3.4. The Constitutive Finite Element Model

The finite element model included concrete pavement, HMA pavement, and base layers from top to bottom. While creating the contact surfaces between the volumes, the contact between the HMA pavement and the concrete pavement was considered standard. The friction coefficient between the layers was regarded as zero to reflect the unbonded condition [13,41]. The friction coefficient between the HMA pavement and the base layer was 0.55 [42]. The gravitational acceleration assigned to the model was 9.81 m/s$^2$ to load the concrete pavement with its weight and flexural strength under temperature differences.

Solid65 was used as the concrete pavement element type. Solid186, which reflects the granular material better, was used for both the HMA and the base layer. To better see the amount of displacement as a boundary condition, the freedom was restricted at the bottom of the model, taking the field support under the base layer, and excluding the base from the calculation [30]. While no restrictions were applied on the model's right side, symmetry restrictions were applied on the left side. The short sides of the model were restricted only in the x direction (longitudinal) to reflect the transverse joint situation. Boundary conditions are presented in Figure 8. As shown in Figure 9, the mesh size of the model was determined by conducting trial studies and using the smart size option. The steps of the finite element analyses are presented in Figure 10.

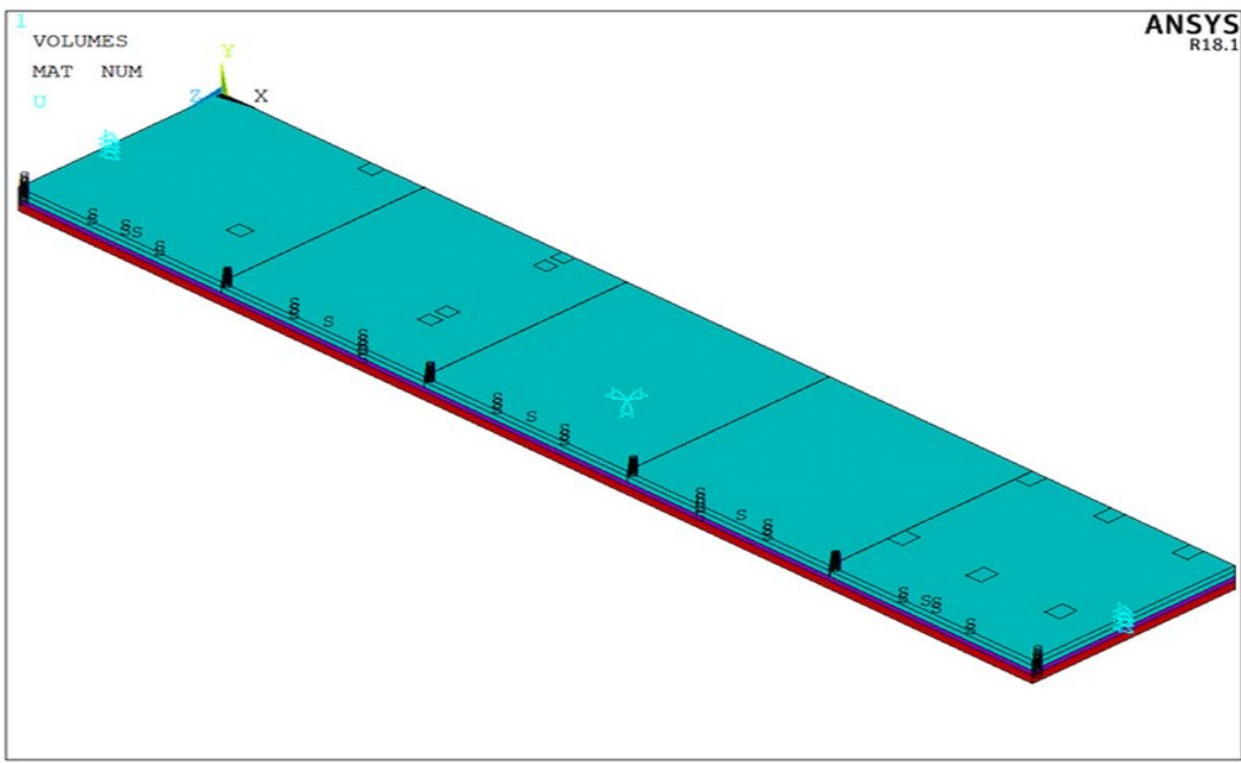

**Figure 8.** Model boundary conditions (ANSYS R18.1).

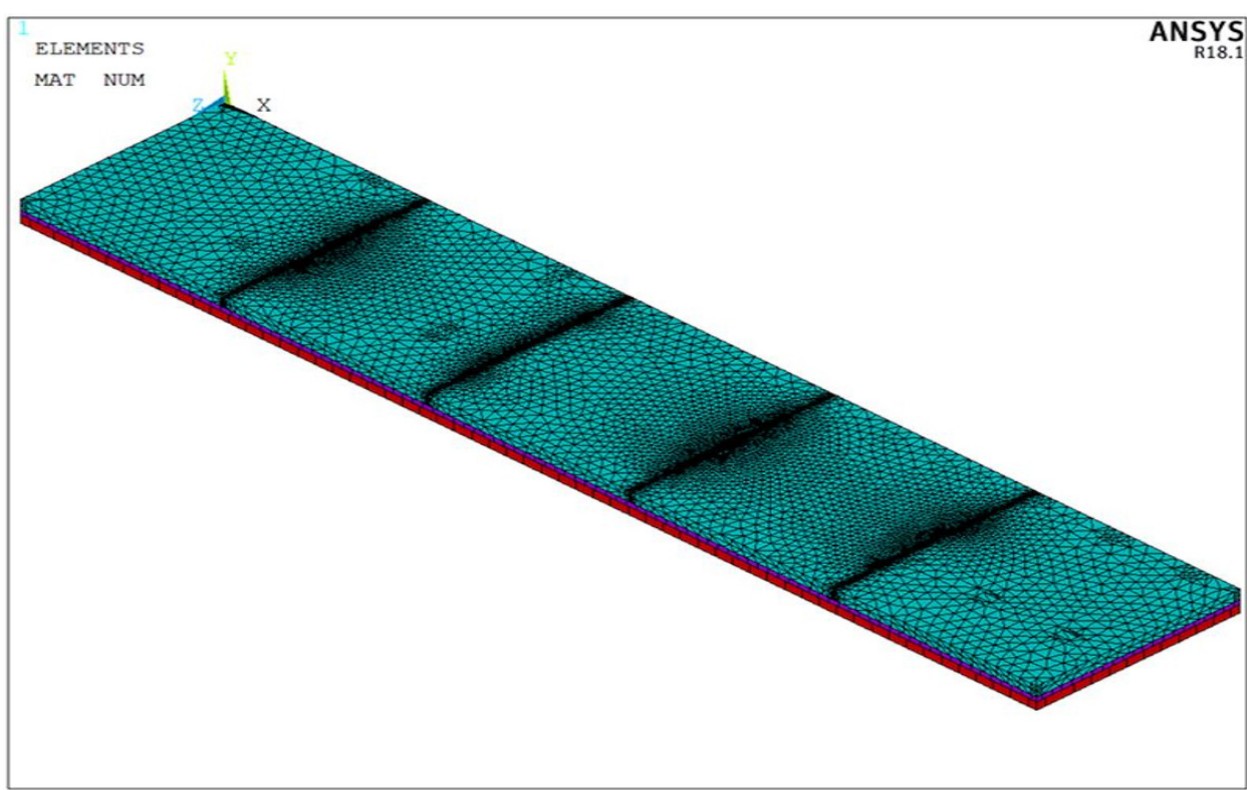

**Figure 9.** Model mesh image (ANSYS R18.1).

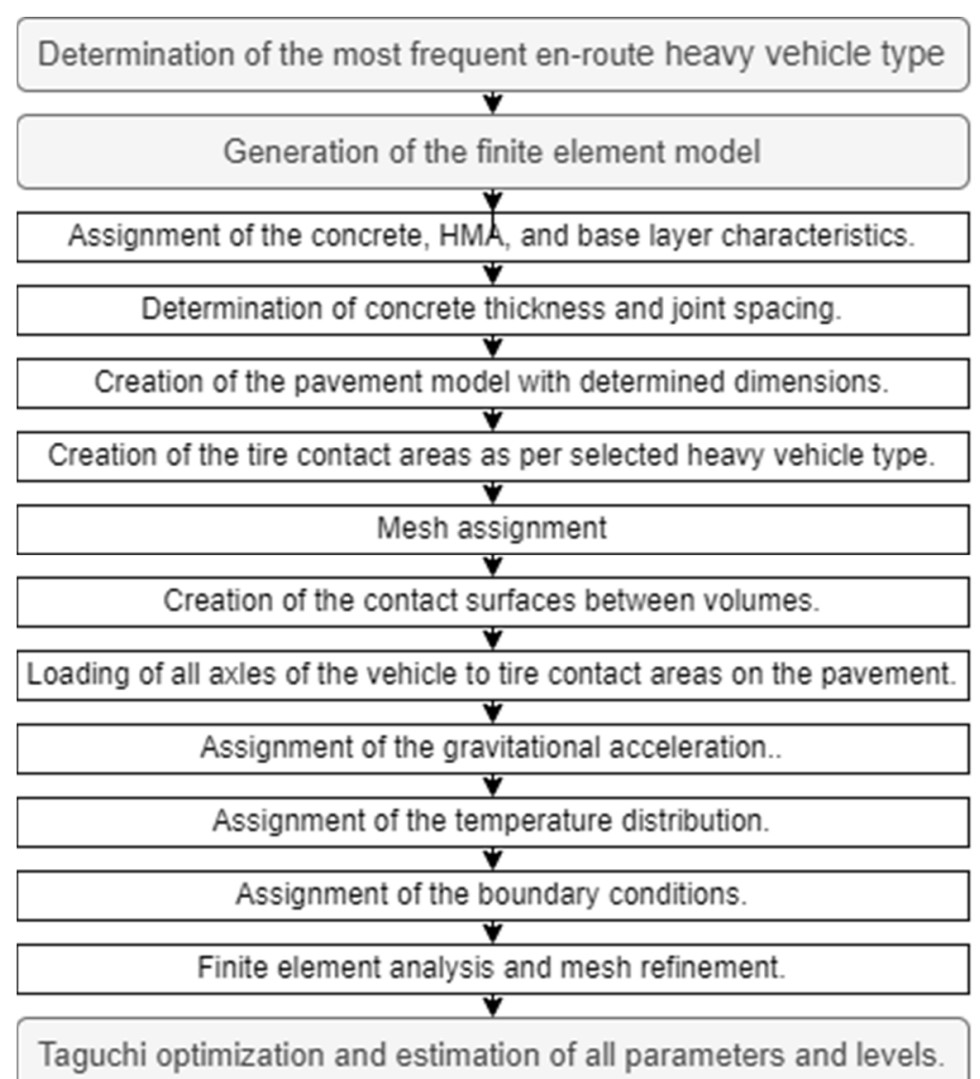

**Figure 10.** Steps of the finite element analyses.

*2.4. Taguchi Method*

Finite element analyses were performed using the Taguchi method, with an L9 orthogonal array. The estimations were carried out with a 95% confidence level using the performance statistics from the calculated stresses. The parameters and levels used in Taguchi optimization are presented in Table 5. The L9 orthogonal array is shown in Table 6.

**Table 5.** The parameters and levels used in the Taguchi method.

| Parameter | Level | | |
|---|---|---|---|
| | 1 | 2 | 3 |
| Slab Thickness (m) | 0.10 | 0.15 | 0.20 |
| Transverse Joint Spacing (m) | 3.50 | 5.00 | 7.00 |
| Modulus of Elasticity of Concrete (MPa) | 34,000 | 36,000 | 37,000 |
| Modulus of Elasticity of HMA (MPa) | 690 | 1379 | 2068 |

**Table 6.** L9 orthogonal array.

| | 4 Parameter 3 Level | | | |
|---|---|---|---|---|
| Experiment No | Slab Thickness (m) | Transverse Joint Spacing (m) | Modulus of Elasticity of Concrete (MPa) | Modulus of Elasticity of HMA (MPa) |
| 1 | 1 | 1 | 1 | 1 |
| 2 | 1 | 2 | 2 | 2 |
| 3 | 1 | 3 | 3 | 3 |
| 4 | 2 | 1 | 2 | 3 |
| 5 | 2 | 2 | 3 | 1 |
| 6 | 2 | 3 | 1 | 2 |
| 7 | 3 | 1 | 3 | 2 |
| 8 | 3 | 2 | 1 | 3 |
| 9 | 3 | 3 | 2 | 1 |

In Taguchi optimization, the most critical loading condition for the concrete slab was loaded together with four different whole configuration axle types. The analyses were performed according to the Taguchi optimization method using bilinear temperature distributions and the slab thickness for four additional whole configuration axle type loadings.

## 3. Results and Discussion

### 3.1. Bilinear Temperature Change in Concrete Slab

In contrast to Westergaard's assumption of a linear temperature distribution, temperature profiles measured at the depth of concrete slabs are not linear [43,44]. Choubane and Tia [45] have suggested considering the nonlinear temperature distribution along with the depth of the concrete slab.

Regarding the literature studies on the temperature distribution along with the depth in concrete slabs, the nonlinear temperature change was calculated with the bilinear change, as shown in Figure 11. To calculate the temperature values in the middle of the concrete slab under positive and negative temperature differences, Formula 1 and 2 were suggested to be used, respectively.

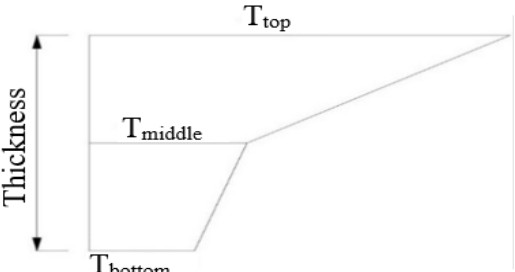

**Figure 11.** Bilinear temperature change in concrete slab.

For positive temperature differences, the temperature in the middle of the concrete slab was calculated as follows:

$$T_{middle} = T_{bottom} + 1/2.93 \ (T_{top} - T_{bottom}). \tag{1}$$

For negative temperature differences, the temperature in the middle of the concrete slab was calculated as follows:

$$T_{middle} = T_{top} + 1/1.76 \ (T_{bottom} - T_{top}). \tag{2}$$

Table 7 presents the temperatures measured along with the pavement depth in the studies in the literature and the temperatures calculated as bilinear. As a result of the regression analysis, $R^2$ was 0.99.

**Table 7.** Comparing the temperatures measured along with the depth of the pavement with the temperatures calculated on a bilinear basis.

| Author | Temperature Differences | Thickness (mm) | Measured Temperature (°C) | Calculated Temperature (°C) |
|---|---|---|---|---|
| Armaghani et al. [46] | Negative | 64 | 18.1 | 18.2 |
| | | 114 | 19.9 | 20.0 |
| | | 165 | 21.7 | 21.3 |
| | Positive | 64 | 32.6 | 32.8 |
| | | 114 | 29.0 | 29.0 |
| | | 165 | 26.4 | 27.0 |
| Karunarathne et al. [47] | Negative | 25 | 34.4 | 34.5 |
| | | 50 | 34.8 | 34.9 |
| | | 75 | 35.4 | 35.3 |
| | | 100 | 35.8 | 36.0 |
| | | 125 | 36.2 | 36.6 |
| | Positive | 25 | 44.6 | 44.0 |
| | | 50 | 44.0 | 42.8 |
| | | 75 | 43.0 | 41.6 |
| | | 100 | 41.9 | 40.4 |
| | | 125 | 40.6 | 39.1 |
| Yu et al. [48] | Negative | 118 | 18.2 | 18.1 |
| | | 177 | 20.2 | 20.0 |
| | | 236 | 21.6 | 21.4 |
| | Positive | 118 | 32.3 | 32.4 |
| | | 177 | 28.0 | 28.2 |
| | | 236 | 25.2 | 26.0 |
| Siddique et al. [49] | Negative | 76 | 27.5 | 27.4 |
| | | 152 | 29.0 | 28.8 |
| | | 229 | 31.0 | 29.9 |
| | Positive | 76 | 36.5 | 38.0 |
| | | 152 | 33.5 | 34.6 |
| | | 229 | 32.0 | 32.8 |
| Kim and Nam [50] | Negative | 70 | 30.1 | 30.1 |
| | | 119 | 31.1 | 30.7 |
| | | 172 | 31.6 | 31.0 |
| | | 224 | 32.1 | 31.7 |
| | | 274 | 32.1 | 32.5 |
| | Positive | 70 | 39.0 | 39.4 |
| | | 119 | 37.1 | 36.7 |
| | | 172 | 35.5 | 35.5 |
| | | 224 | 34.0 | 33.1 |
| | | 274 | 32.5 | 30.8 |
| Choubane and Tia [45] | Negative | 64 | 28.9 | 28.9 |
| | | 114 | 30.4 | 30.2 |
| | | 165 | 31.5 | 31.2 |
| | Positive | 64 | 43.8 | 43.8 |
| | | 114 | 39.4 | 39.6 |
| | | 165 | 36.8 | 37.5 |

When the concrete slab has a positive temperature difference, the critical load location for transverse cracks is at the transverse joint edge and the corner of the slab [51]. For

that reason, positive temperature differences were used in this study. The maximum positive temperature differences and linear-gradient values between the top and bottom of the pavement measured in the studies carried out to monitor the temperature changes in concrete pavements are presented in Table 8. The average linear-gradient value of 0.52 °C/cm and the thermal expansion coefficient of $9 \times 10^{-6}$/°C were regarded as the linear gradient value. According to this value, the temperature difference between the top and bottom of the concrete pavement was calculated using the suggested formulas. As described previously, the temperature changes throughout the pavement thickness were calculated as bilinear.

**Table 8.** Maximum positive and negative temperature differences in literature studies.

| Previous Studies | Thickness (cm) | Max. Pos. Temp. Difference °C | Max. Pos. Temp. Linear Gradient °C/cm | Max. Neg. Temp. Difference °C | Study Area | Measurement Time for Pos. Temp. Difference | Measurement Time for Neg. Temp. Difference |
|---|---|---|---|---|---|---|---|
| Armaghani and Larsen [46] | 23 | 11.0 | 0.48 | −6.1 | Site | Summer | Summer |
| Richardson and Armaghani [52] Shoukry and Fahmy [53] | 22.5 | 10.0 | 0.44 | | Site | Summer | |
| Choubane and Tia [45] | 20.3 | 12.7 | 0.63 | −4.6 | Site | Summer | Summer |
| Byrum and Hansen [54] | 17.8 | 15.5–19.4 | 0.87–1.09 | −7.8 to −11.6 | Site | Summer | |
| Yu et al. [48] | 29.5 | 12.8 | 0.43 | −6.7 | Site | Summer | Summer |
| Siddique et al. [49] | 30.5 | 10.5 | 0.34 | −5.0 | Site | Summer | Summer |
| Karunarathne et al. [47] | 15 | 7.9 | 0.53 | −2.6 | Site | Summer | Summer |
| Kim and Nam [50] | 32.6 | 10.0 | 0.31 | −2.5 | Site | Summer | Summer |
| Maitra et al. [55] | 30 | 26.8 | 0.89 | | | Laboratory Conditions | |
| Salman and Patil [15] | 20.32 | 13.4 | 0.66 | −6.7 | | Finite Element Analysis | |
| Dhananjay and Abhilash [56] | 25 | 13.2 | 0.53 | −3.5 | Site | Summer | Summer |
| Kim and Chun [57] | 23 | 5.0 | 0.22 | −4.4 | Site | Winter | Winter |
| Pancar and Akpınar [58] | 25 | 14.0 | 0.56 | −9.8 | Site | Summer | Winter |
| Kamalakara et al. [59] | 30 | 15.0 | 0.5 | −6.0 | Site | Summer | Summer |

The temperature distributions were calculated along with the pavement thickness and assigned to the ANSYS finite element models for concrete pavements with 100 mm, 150 mm, and 200 mm thicknesses.

*3.2. 3-Axle Truck Loading Analysis Results*

For three-axle (1.22) heavy vehicle loads, the maximum principal stresses and displacements occurring under the concrete slab and on the transverse and longitudinal joint edges due to temperature differences were found in the finite element analysis. Maximum stresses under the longitudinal joint were also calculated with the Bradbury formulas. Maximum principal stresses and displacements under the concrete slab induced by temperature differences and whole configuration axle-type loading were found in the finite element analyses. The results are presented in Table 9.

In the finite element analysis, it was noticed that the optimum levels (the levels that minimized the maximum principal stress) according to the factors chosen in the pavement model were A3, B2, C2, and D3. As shown in Table 11, the highest contribution to the S/N parameter indicated the highest contribution upon the maximum principal stress. For that reason, it was determined that the slab thickness had the highest effect on the maximum principal stress. This was followed by the modulus of elasticity of the concrete pavement, the modulus of elasticity of the HMA pavement, and the joint spacing. When the experiment plan shown in Table 6 was analyzed, it was noticed that no design indicated the optimum result. Finite element analyses were conducted to determine the effect in preparing the model according to the relevant parameter levels (A3, B2, C2, D3). As a result of the verification test, the maximum principal stress was found to be 4.13 MPa, as presented in Figure 12. The S/N value corresponding to this result was −12.319. In a 95% confidence interval, this value should be between −12.517 and −11.233. Therefore, it is possible to state that the 95% confidence interval's calculated value was accurate.

**Table 9.** Second model three-axle heavy vehicle loading test results.

| | Second Model Three-Axle Heavy Vehicle Loading | | | | | |
|---|---|---|---|---|---|---|
| | Maximum Principal Stresses and Displacements Induced by Temperature Differences | | | | Maximum Principal Stresses and Displacements Induced by Temperature Differences and Loading | |
| Experiment No | ANSYS | | | Bradbury | ANSYS | |
| | Max Displacement | Transverse Joint | Longitudinal Joint Bottom | Longitudinal Joint Bottom | Max Displacement | Transverse Joint |
| | DMX (mm) | $S1_{MAX}$ (MPa) | S1 (MPa) | S1 (MPa) | DMX (mm) | $S1_{MAX}$ (MPa) |
| 1 | 0.15 | 1.13 | 1.10 | 0.96 | 0.80 | 7.13 |
| 2 | 0.15 | 1.23 | 1.20 | 1.01 | 0.77 | 6.65 |
| 3 | 0.15 | 1.26 | 1.24 | 1.04 | 0.77 | 7.16 |
| 4 | 0.24 | 3.45 | 1.68 | 1.45 | 0.65 | 4.16 |
| 5 | 0.24 | 3.39 | 1.83 | 1.44 | 0.67 | 4.70 |
| 6 | 0.23 | 3.41 | 1.74 | 1.33 | 0.67 | 4.36 |
| 7 | 0.34 | 4.43 | 2.35 | 2.08 | 0.65 | 4.45 |
| 8 | 0.33 | 3.76 | 2.19 | 1.78 | 0.65 | 3.92 |
| 9 | 0.34 | 4.12 | 2.34 | 1.87 | 0.69 | 4.06 |

The average S/N effects obtained using the performance statistics are presented in Table 10. The conditions and performance estimations that minimize the maximum principal stress under the concrete slab using the average S/N effects are shown in Table 11.

**Table 10.** S/N effects table.

| | Average S/N Effects | | | |
|---|---|---|---|---|
| | Slab Thickness | Joint Spacing | Modulus of Elasticity of Concrete | Modulus of Elasticity of HMA |
| 1st Level | −16.872 | −14.137 | −13.906 | −14.225 |
| 2nd Level | −12.871 | −13.921 | −13.670 | −14.071 |
| 3rd Level | −12.334 | −14.020 | −14.502 | −13.782 |

**Table 11.** Performance estimations table.

| Factors | Level | Contribution upon S/N |
|---|---|---|
| Slab Thickness | A3 | 1.692 |
| Joint Spacing | B2 | 0.105 |
| Modulus of Elasticity of Concrete | C2 | 0.356 |
| Modulus of Elasticity of HMA | D3 | 0.244 |
| Contribution of All Factors (S/N) | | 2.397 |
| Average Performance Statistics (S/N) | | −14.026 |
| Expected Value at Optimum Conditions (S/N) | | −11.629 |
| Verification Test Result (S/N)/(MPa) | | −12,320/4.13 |
| Confidence Interval ($\alpha$ = 95%) (S/N) | | −12,517/−11.233 |

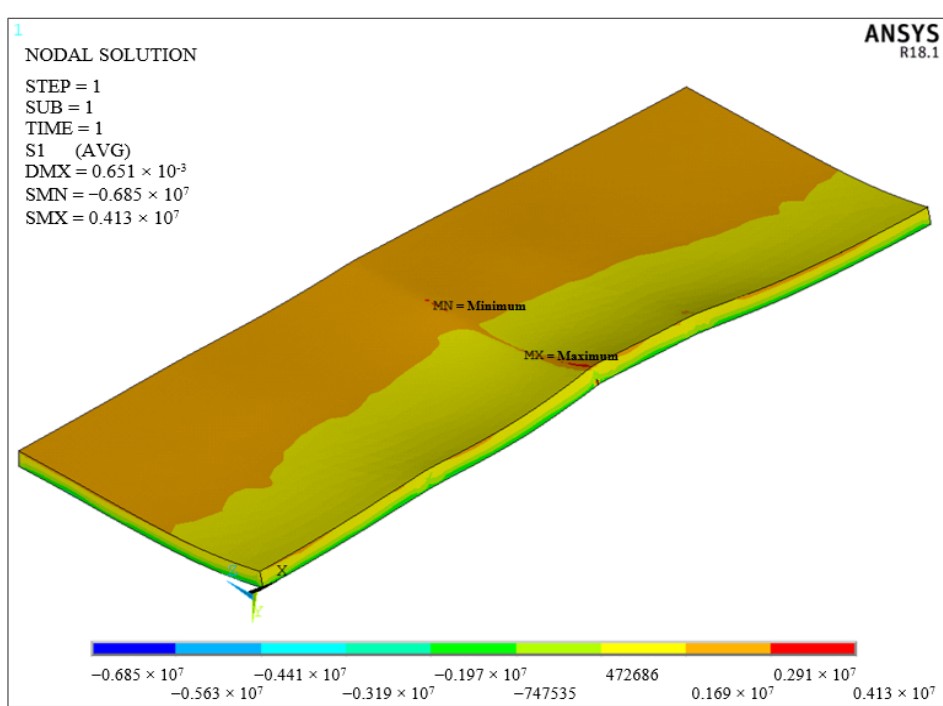

**Figure 12.** Maximum principal stresses arising from whole configuration axle type loading and temperature difference with optimum levels of the three-axle loading condition (ANSYS R18.1).

### 3.3. Evaluation of Three-Axle Truck Loading Analysis Results

When the results were analyzed, it was noticed that the maximum principal stress values that appeared under the concrete slab induced by the temperature difference and whole configuration axle type loading were at the third level of the slab thickness, at the second level of the joint spacing, at the second level of the modulus of elasticity of concrete, and at the third level of the modulus of elasticity of HMA.

According to the literature, the maximum principal stresses under the concrete slab decreased significantly with the thickness increase. As presented in Figure 13, the third level of this parameter was noticed at the 0.20 m concrete pavement thickness.

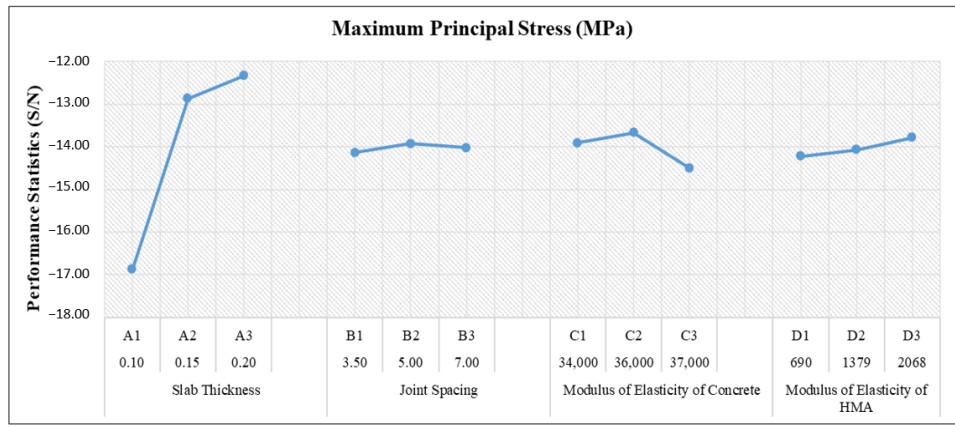

**Figure 13.** Performance statistics for all levels.

It was observed that the effect of the joint spacing on SN was low, with a value of 0.105, and the lowest maximum principal stress was obtained under the concrete slab at the second level. However, in the literature studies, it has been observed that the cracks that occurred in the concrete slab increased with the increment in joint spacing [37].

With the increment in the modulus of elasticity of concrete, the maximum stress values occurring under the slab increase [51]. However, the increase in the modulus of elasticity

of concrete is associated with the increase in the flexural strength of the concrete. It was noticed that the effect of the modulus of elasticity of concrete on SN was low, with a value of 0.356, and the lowest maximum principal stress was obtained under the concrete slab at the second level.

In the bonded concrete overlay design, an increase in the modulus of elasticity of HMA results in a decrease in the stresses occurring under the concrete slab. However, in the unbonded concrete overlay design, the modulus of elasticity of the HMA layer has a lesser effect [13]. It was also specified that the effect of the modulus of elasticity of HMA on S/N was low, with a value of 0.244, and the lowest maximum principal stress was obtained under the concrete slab at the third level.

In light of these resulting data, alternative solutions were proposed to the designers before their final decision, according to the existing deteriorated flexible pavement condition and the most frequent en-route heavy vehicle axle type. The predicted values obtained from the Taguchi method for the maximum principal stress (MPS) of three-axle trucks are presented in Table 12. The tables for the other axle types were presented in [60]. The flow chart to choose the recommended design is shown in Figure 14.

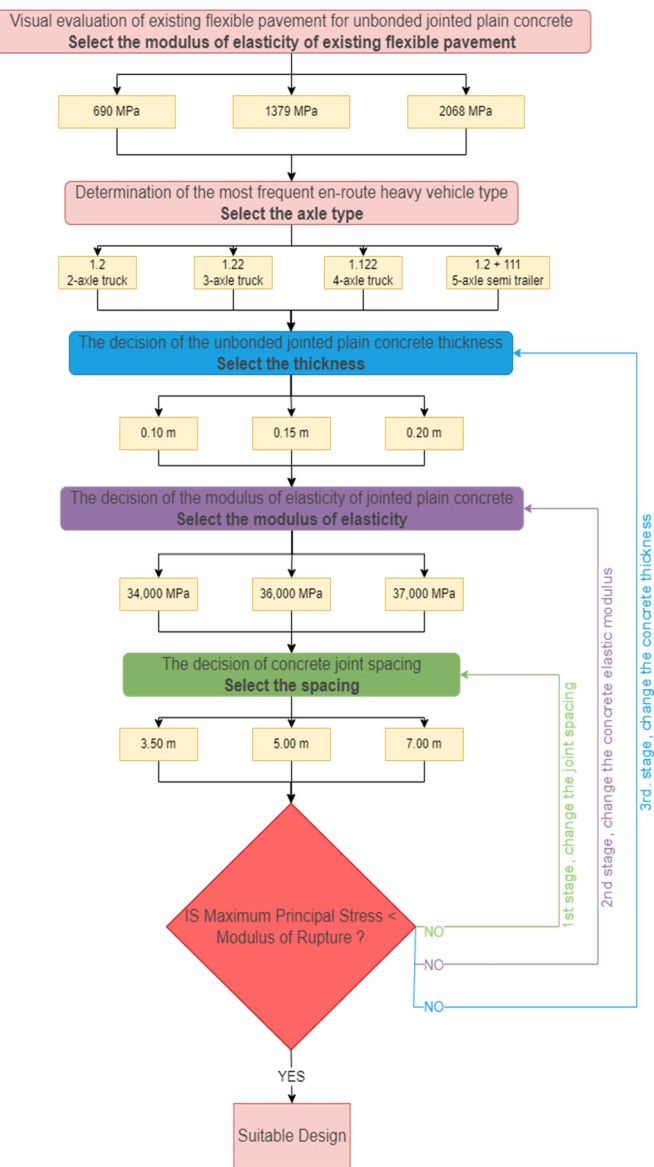

**Figure 14.** Pavement design flow chart.

**Table 12.** Maximum principal stress tables for three-axle type (1.22) vehicles.

| Elasticity Mod. of HMA (MPa) | Heavy Vehicle Axle Type | Concrete Thickness (m) | Elasticity Mod. of Concrete (MPa) | Joint Spacing (m) | Maximum Principal Stress (MPa) | Rupture Mod. of Concrete (MPa) | Result |
|---|---|---|---|---|---|---|---|
| 690 | 1.22 | 0.10 | 34,000 | 3.50 | 7.13 | 4.40 | Not Suitable |
| 690 | 1.22 | 0.10 | 34,000 | 5.00 | 6.96 | 4.40 | Not Suitable |
| 690 | 1.22 | 0.10 | 34,000 | 7.00 | 7.03 | 4.40 | Not Suitable |
| 690 | 1.22 | 0.10 | 36,000 | 3.50 | 6.94 | 4.60 | Not Suitable |
| 690 | 1.22 | 0.10 | 36,000 | 5.00 | 6.77 | 4.60 | Not Suitable |
| 690 | 1.22 | 0.10 | 36,000 | 7.00 | 6.85 | 4.60 | Not Suitable |
| 690 | 1.22 | 0.10 | 37,000 | 3.50 | 7.64 | 5.00 | Not Suitable |
| 690 | 1.22 | 0.10 | 37,000 | 5.00 | 7.45 | 5.00 | Not Suitable |
| 690 | 1.22 | 0.10 | 37,000 | 7.00 | 7.53 | 5.00 | Not Suitable |
| 690 | 1.22 | 0.15 | 34,000 | 3.50 | 4.39 | 4.40 | Suitable |
| 690 | 1.22 | 0.15 | 34,000 | 5.00 | 4.39 | 4.40 | Suitable |
| 690 | 1.22 | 0.15 | 34,000 | 7.00 | 4.38 | 4.40 | Suitable |
| 690 | 1.22 | 0.15 | 36,000 | 3.50 | 4.38 | 4.60 | Suitable |
| 690 | 1.22 | 0.15 | 36,000 | 5.00 | 4.27 | 4.60 | Suitable |
| 690 | 1.22 | 0.15 | 36,000 | 7.00 | 4.32 | 4.60 | Suitable |
| 690 | 1.22 | 0.15 | 37,000 | 3.50 | 4.82 | 5.00 | Suitable |
| 690 | 1.22 | 0.15 | 37,000 | 5.00 | 4.70 | 5.00 | Suitable |
| 690 | 1.22 | 0.15 | 37,000 | 7,00 | 4.75 | 5.00 | Suitable |
| 690 | 1.22 | 0.20 | 34,000 | 3.50 | 4.23 | 4.40 | Suitable |
| 690 | 1.22 | 0.20 | 34,000 | 5.00 | 4.12 | 4.40 | Suitable |
| 690 | 1.22 | 0.20 | 34,000 | 7.00 | 4.17 | 4.40 | Suitable |
| 690 | 1.22 | 0.20 | 36,000 | 3.50 | 4.12 | 4.60 | Suitable |
| 690 | 1.22 | 0.20 | 36,000 | 5.00 | 4.01 | 4.60 | Suitable |
| 690 | 1.22 | 0.20 | 36,000 | 7.00 | 4.06 | 4.60 | Suitable |
| 690 | 1.22 | 0.20 | 37,000 | 3.50 | 4.53 | 5.00 | Suitable |
| 690 | 1.22 | 0.20 | 37,000 | 5.00 | 4.42 | 5.00 | Suitable |
| 690 | 1.22 | 0.20 | 37,000 | 7.00 | 4.47 | 5.00 | Suitable |
| 1379 | 1.22 | 0.10 | 34,000 | 3.50 | 7.00 | 4.40 | Not Suitable |
| 1379 | 1.22 | 0.10 | 34,000 | 5.00 | 6.83 | 4.40 | Not Suitable |
| 1379 | 1.22 | 0.10 | 34,000 | 7.00 | 6.91 | 4.40 | Not Suitable |
| 1379 | 1.22 | 0.10 | 36,000 | 3.50 | 6.82 | 4.60 | Not Suitable |
| 1379 | 1.22 | 0.10 | 36,000 | 5.00 | 6.65 | 4.60 | Not Suitable |
| 1379 | 1.22 | 0.10 | 36,000 | 7.00 | 6.73 | 4.60 | Not Suitable |
| 1379 | 1.22 | 0.10 | 37,000 | 3.50 | 7.50 | 5.00 | Not Suitable |
| 1379 | 1.22 | 0.10 | 37,000 | 5.00 | 7.32 | 5.00 | Not Suitable |
| 1379 | 1.22 | 0.10 | 37,000 | 7.00 | 7.40 | 5.00 | Not Suitable |
| 1379 | 1.22 | 0.15 | 34,000 | 3.50 | 4.35 | 4.40 | Suitable |
| 1379 | 1.22 | 0.15 | 34,000 | 5.00 | 4.31 | 4.40 | Suitable |
| 1379 | 1.22 | 0.15 | 34,000 | 7.00 | 4.36 | 4.40 | Suitable |
| 1379 | 1.22 | 0.15 | 36,000 | 3.50 | 4.30 | 4.60 | Suitable |
| 1379 | 1.22 | 0.15 | 36,000 | 5.00 | 4.20 | 4.60 | Suitable |
| 1379 | 1.22 | 0.15 | 36,000 | 7.00 | 4.24 | 4.60 | Suitable |
| 1379 | 1.22 | 0.15 | 37,000 | 3.50 | 4.73 | 5.00 | Suitable |
| 1379 | 1.22 | 0.15 | 37,000 | 5.00 | 4.62 | 5.00 | Suitable |
| 1379 | 1.22 | 0.15 | 37,000 | 7.00 | 4.67 | 5.00 | Suitable |
| 1379 | 1.22 | 0.20 | 34,000 | 3.50 | 4.15 | 4.40 | Suitable |
| 1379 | 1.22 | 0.20 | 34,000 | 5.00 | 4.05 | 4.40 | Suitable |
| 1379 | 1.22 | 0.20 | 34,000 | 7.00 | 4.10 | 4.40 | Suitable |
| 1379 | 1.22 | 0.20 | 36,000 | 3.50 | 4.04 | 4.60 | Suitable |
| 1379 | 1.22 | 0.20 | 36,000 | 5.00 | 3.94 | 4.60 | Suitable |
| 1379 | 1.22 | 0.20 | 36,000 | 7.00 | 3.99 | 4.60 | Suitable |
| 1379 | 1.22 | 0.20 | 37,000 | 3.50 | 4.45 | 5.00 | Suitable |
| 1379 | 1.22 | 0.20 | 37,000 | 5.00 | 4.34 | 5.00 | Suitable |
| 1379 | 1.22 | 0.20 | 37,000 | 7.00 | 4.39 | 5.00 | Suitable |

## 4. Conclusions

In this study, the concrete class, slab thickness, and sizes ensuring that the maximum principal stress occurring under the concrete pavement remains below the rupture modulus of the concrete for an unbonded concrete pavement design, to be applied to an HMA pavement subjected to structural and functional deterioration, were determined with 3D non-linear finite element analysis (ANSYS) and compared with the AASHTO method. The most critical loading conditions on the slab were used: edge and joint loading, whole configuration axle-type loading, and bilinear temperature distributions along with the concrete slab thickness. The worst conditions were considered in determining the thickness and material properties of the pavement layers according to the AASHTO design method. Because the factors affecting the pavement design and the solutions to be found depending on these factors were high, finite element analyses were carried out according to the Taguchi method as a unique experimental design technique. Heavy vehicles with axle types 1.2, 1.22, 1.122, and 1.2 + 111, as the most common on our country's highways, were used. The results of this study are as follows.

The critical axle type was noticed to be 1.2 in the optimizations. It was observed that a 0.15 m concrete slab thickness was sufficient for the 1.22, 1.122, and 1.2 + 111 axle types, but not for the 1.2 axle types. For the 1.2 axle type, the minimum concrete slab thickness was 0.20 m, and the concrete classes to be used were C45/55 and C50/60. This revealed that the design should be planned with whole configuration axle loading instead of single-, tandem-, or tridem-axle loading in pavement design.

The required concrete overlay thicknesses for concrete classes C40/50, C45/55, and C50/60 were calculated to be 0.222 m, 0.217 m, and 0.206 m, respectively, using the AASHTO method. In the optimizations administered for the critical 1.2 axle-type loading, it was observed that the maximum principal stress that appeared under the slab for the C40/50 concrete class was above the concrete rupture modulus. Therefore, the minimum concrete thickness required for concrete types C45/55 and C50/60 was noticed to be 0.20 m. Slab thicknesses close to each other were found with the AASHTO and finite element methods.

Under the selected loading parameters, it was recommended that, at minimum, the C45/55 class and a 0.20 m-thick unbonded unreinforced concrete overlay should be constructed on HMA pavements in fair and deteriorated conditions to improve and extend the service life.

The maximum principal stresses that appeared under the slab due to the temperature alone were more critical for concrete slabs with a thickness of 0.20 m, under the selected loading parameters.

It was observed that the maximum displacement remained approximately the same when the concrete slab thickness was kept constant, and the material properties of the concrete, asphalt, and foundation layers and the transverse joint spacing were changed. Due to this, it was considered that the maximum displacement due to temperature differences was mainly associated with the concrete slab thickness.

In the optimizations, the slab thickness was the most efficient parameter concerning the maximum principal stresses under the concrete slab. It was observed that the slab length had little effect on the maximum principal stress. However, it should be kept in mind that the cracks at the joint spacing also increased. The effect of the modulus of elasticity of HMA pavement on the maximum principal stress was more negligible due to choosing the unbonded design method.

The variations in HMA, subbase, and base properties were found to have minimal effect on the maximum principal stresses induced under the concrete pavement. Therefore, the unbonded concrete pavement design was considered suitable for many types of HMA, subbases, and bases.

In this study, based on rural roads, dowel bars in the transverse joints and tie bars in the longitudinal joints were not used. In future studies, pavement designs can be established for urban roads by using dowel bars, tie bars, and the full configuration axle type, and

by calculating different temperature distributions throughout the concrete slab thickness with the suggested formulas. Experimental studies can be conducted to validate the finite element analyses.

**Author Contributions:** Conceptualization, F.İ.B. and O.Ü.B.; methodology, F.İ.B. and O.Ü.B.; software, F.İ.B. and H.F.B.; validation, F.İ.B., O.Ü.B. and H.F.B.; formal analysis, F.İ.B.; investigation, F.İ.B. and H.F.B.; resources, F.İ.B.; data curation, H.F.B.; writing—original draft preparation, F.İ.B.; writing—review and editing, F.İ.B., O.Ü.B. and H.F.B.; visualization, F.İ.B.; supervision, O.Ü.B. All authors have read and agreed to the published version of the manuscript.

**Funding:** This research received no external funding.

**Institutional Review Board Statement:** Not applicable.

**Informed Consent Statement:** Not applicable.

**Data Availability Statement:** Not applicable.

**Conflicts of Interest:** The authors declare no conflict of interest.

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
