# Peer review of "A Practical Design Guide for Unbonded Jointed Plain Concrete Roads over Deteriorated HMA Roads: Realistic Traffic Loading"

_coatings, doi:10.3390/coatings12121817_

Round 1

Reviewer 1 Report

The following points could be addressed to further improve the manuscript quality.

(1) In part 2.3.2, it is better to explain a bit more about the inflation pressure of all tires was set as 700 kPa.

(2) In part 2.3.3, no Poisson ratio value was given to reflect the worst case.

(3) The modulus of elasticity of the existing asphalt layer was taken to be 690, 1379, and 2068 MPa. Please explain what state the asphalt layer with elastic modulus of 1379MPa and 2068MPA represents.

(4) Table 10, what about the average S/N effects, especially how to calculate it.

(5) In part 3.3, the effect of the joint spacing on SN was low, with the value of 0.105, and what are possible reasons?

(6) Figure 13 shows the pavement design flow chart. It is suggested that the specific steps of pavement design in detail be given.

(7) The couclusion section is too long, and it should be concluded and more concise.

Author Response

Authors are thankful to Reviewer truly because of their valuable comments.  We did our best in order to complete the deficiencies of our article with respect to your opinions. Please see the attachment.

Reviewer 2 Report

This paper conducts numerical research on determining the material properties and the thickness of the layers for the unbonded jointed plain concrete pavement design over the deteriorated flexible pavement. The paper is, in general attractive, and it needs minor improvement before it can be recommended for publication. The following comments must be addressed in the next version of the paper.

(1) The conclusion's content is too much and needs to be simplified.

(2) What is "1.2+111" in Fig.13? Can each variable data be taken as a range rather than a specific value?

(3) Fig. 14 is too vague. It is recommended to make a table.

(4) In the future, it is recommended to use experiments for analysis.

Author Response

(The authors gave the same response as above.)

Reviewer 3 Report

This manuscript examined the Unbonded Jointed Plain Concrete
Roads over Deteriorated HMA Roads.
The manuscript needs thorough corrections based on the following comments:

1. There are many sentences in the text that have errors in grammar and should be corrected. The authors suggest doing a proof of English reading and editing a manuscript to correct all grammar errors.

2. What is the significance of the research? Please add a research significance section.

3. It is preferable to add a section "Recommendations" to indicate the authors' recommendations for future researchers.

4. Conclusions need more improvement.

5. The research introduction needs more attention than the current situation to show the gap between what has been studied and the importance of the current research.

6. More recent references should be added to the introduction such as:
1.Numerical analysis of the shear behavior of FRP-strengthened continuous RC beams having web openings.

2. Effectiveness of SHCC strips reinforced with glass fiber textile mesh layers for shear strengthening of RC beams: Experimental and numerical assessments.

3.Behavior of RC beams strengthened in shear with ultra-high performance fiber reinforced concrete (UHPFRC).

4.Behavior of steel I-beam embedded in normal and steel fiber reinforced concrete incorporating demountable bolted connectors.

5. Bond behavior between concrete and prefabricated Ultra High-Performance Fiber-Reinforced Concrete (UHPFRC) plates.

6.Effect of interfacial surface preparation technique on bond characteristics of both NSC-UHPFRC and NSC-NSC composites.

7.Flexural strengthening of RC one way solid slab with Strain Hardening Cementitious Composites (SHCC).

8. Finite element analysis of shear performance of UHPFRC-encased steel composite beams: Parametric study.

9.Evaluate the effect of steel, polypropylene and recycled plastic fibers on concrete properties.

10. Flexural rigidity and ductility of RC beams reinforced with steel and recycled plastic fibers

11. Finite element analysis of shear performance of UHPFRC-encased steel composite beams: Parametric study

12. Shear behavior of RC pile cap beams strengthened using ultra-high performance concrete reinforced with steel mesh fabric

7. How can the results of this paper help to improve the specification?

8.Quality of all figures and tables should be improved, there were many data cannot be easily read.

9. The constitutive model used in FEM should be explained in detail.

10. What is the effect of the mesh size on the model efficiency?.

Author Response

(The authors gave the same response as above.)

Reviewer 4 Report

The manuscript is an interesting study and I really enjoyed reading the paper. However, in my opinion, the manuscript has some shortcomings in the text. According to the mentioned items, I recommend minor revision for the manuscript.

1.      The abstract encounters a lack of quantitative values. The author should revise the abstract again regarding this issue.

2.      Although the literature review is very important, the authors haven’t presented this section. Therefore, it is recommended to add this section at the end of the introduction section with more details.

3.      Some figures do not have suitable quality and authors should substitute them with new ones.

4.      It is better to present a flow chart in the material and methods section.

5.      Most of the references are outdated. New papers (2021 and 2022) must be added to the paper.

6.      What is the main question addressed by the research?

7.      Do you consider the topic original or relevant in the field? Does it address a specific gap in the field?

8.      What does it add to the subject area compared with other published
material?

9.      What specific improvements should the authors consider regarding the methodology?

10.    What further controls should be considered?

11.   Are the conclusions consistent with the evidence and arguments presented and do they address the main question posed?

12.   Are the references appropriate?

13.   Please include any additional comments on the tables and figures.

Best Regards,

Author Response

(The authors gave the same response as above.)

Round 2

Reviewer 1 Report

The authors addressed my problems and it is suggested to be accepted now.

Reviewer 3 Report

The authors adequately revised the paper.